# Zeroth-Order Negative Curvature Finding: Escaping Saddle Points without Gradients

**Hualin Zhang**[1]    **Huan Xiong**[2,3]    **Bin Gu**[1,3]

[1] Nanjing University of Information Science & Technology
[2] Harbin Institute of Technology
[3] Mohamed bin Zayed University of Artificial Intelligence
{zhanghualin98,huan.xiong.math,jsgubin}@gmail.com

## Abstract

We consider escaping saddle points of nonconvex problems where only the function evaluations can be accessed. Although a variety of works have been proposed, the majority of them require either second or first-order information, and only a few of them have exploited zeroth-order methods, particularly the technique of negative curvature finding with zeroth-order methods which has been proven to be the most efficient method for escaping saddle points. To fill this gap, in this paper, we propose two zeroth-order negative curvature finding frameworks that can replace Hessian-vector product computations without increasing the iteration complexity. We apply the proposed frameworks to ZO-GD, ZO-SGD, ZO-SCSG, ZO-SPIDER and prove that these ZO algorithms can converge to $(\epsilon, \delta)$-approximate second-order stationary points with less query complexity compared with prior zeroth-order works for finding local minima.

## 1 Introduction

Nonconvex optimization has received wide attention in recent years due to its popularity in modern machine learning (ML) and deep learning (DL) tasks. Specifically, in this paper, we study the following unconstrained optimization problem:

$$\min_{x \in \mathbb{R}^d} f(x) := \frac{1}{n} \sum_{i=1}^{n} f_i(x), \tag{1}$$

where both $f_i(\cdot)$ and $f(\cdot)$ can be nonconvex. In general, finding the global optima of nonconvex functions is NP-hard. Fortunately, finding local optima is an alternative because it has been shown in theory and practice that local optima have comparable performance capabilities to global optima in many machine learning problems [14, 15, 26, 17, 16, 19, 27]. Gradient-based methods have been shown to be able to find an $\epsilon$-approximate first-order stationary point ($\|\nabla f(x)\| \leq \epsilon$) efficiently, both in the deterministic setting (*e.g.*, gradient descent [32]; accelerated gradient descent [8, 29]) and stochastic setting (*e.g.*, stochastic gradient descent [32, 37]; SCSG [28]; SPIDER [13]). However, in nonconvex settings, first-order stationary points can be local minima, global minima, or even saddle points. Converging to saddle points will lead to highly suboptimal solutions [20, 39] and destroy the model's performance. Thus, escaping saddle points has recently become an important research topic in nonconvex optimization.

Several classical results have shown that, for $\rho$-Hessian Lipschitz functions (see Definition 1), using the second-order information like computing the Hessian [33] or Hessian-vector products [1, 9, 2], one can find an $\epsilon$-approximate second-order stationary point (SOSP, $\|\nabla f(x)\| \leq \epsilon$ and $\nabla^2 f(x) \succeq -\sqrt{\rho\epsilon}\mathbf{I}$). However, when the dimension of $x$ is large, even once access to the Hessian

36th Conference on Neural Information Processing Systems (NeurIPS 2022).

is computationally infeasible. A recent line of work shows that, by adding uniform random perturbations, first-order (FO) methods can efficiently escape saddle points and converge to SOSP. In the deterministic setting, [22] proposed the perturbed gradient descent (PGD) algorithm with gradient query complexity $\tilde{\mathcal{O}}(\log^4 d/\epsilon^2)$ by adding uniform random perturbation into the standard gradient descent algorithm. This complexity is later improved to $\tilde{\mathcal{O}}(\log^6 d/\epsilon^{1.75})$ by the perturbed accelerated gradient descent [24] which replaces the gradient descent step in PGD by Nesterov's accelerated gradient descent.

Table 1: A summary of the results of finding $(\epsilon, \delta)$-approximate SOSPs (see Definition 2) by the zeroth-order algorithms. (CoordGE, GaussGE, and RandGE are abbreviations of "coordinate-wise gradient estimator", "Gaussian random gradient estimator" and "uniform random gradient estimator", respectively. RP, RS, and CR are abbreviations of "random perturbation", "random search" and "cubic regularization", respectively.)

| Algorithm | Setting | ZO Oracle | Main Techniques | Function Queries |
|---|---|---|---|---|
| ZPSGD [23] | Deterministic | GaussGE + Noise | RP | $\tilde{\mathcal{O}}\left(\frac{d^2}{\epsilon^5}\right)$ † |
| PAGD [41] | Deterministic | CoordGE | RP | $\mathcal{O}\left(\frac{d\log^4 d}{\epsilon^2}\right)$ † |
| RSPI [30] | Deterministic | CoordGE | RS + NCF | $\mathcal{O}(\frac{d\log d}{\epsilon^{8/3}})$ ‡ |
| **Theorem. 4** | Deterministic | CoordGE | NCF | $\mathcal{O}\left(\frac{d}{\epsilon^2} + \frac{d\log d}{\delta^{3.5}}\right)$ |
| ZO-SCRN [5] | Stochastic | GaussGE | CR | $\tilde{\mathcal{O}}\left(\frac{d}{\epsilon^{3.5}} + \frac{d^4}{\epsilon^{2.5}}\right)$ † |
| **Theorem. 3** | Stochastic | CoordGE | NCF | $\tilde{\mathcal{O}}\left(\frac{d}{\epsilon^4} + \frac{d}{\epsilon^2\delta^3} + \frac{d}{\delta^5}\right)$ |
| **Theorem. 5** | Stochastic | CoordGE + (RandGE) | NCF | $\tilde{\mathcal{O}}\left(\frac{d}{\epsilon^{10/3}} + \frac{d}{\epsilon^2\delta^3} + \frac{d}{\delta^5}\right)$ |
| **Theorem. 6** | Stochastic | CoordGE | NCF | $\tilde{\mathcal{O}}\left(\frac{d}{\epsilon^3} + \frac{d}{\epsilon^2\delta^2} + \frac{d}{\delta^5}\right)$ |

† guarantees $(\epsilon, \mathcal{O}(\sqrt{\epsilon}))$-approximate SOSP, and ‡ guarantees $(\epsilon, \epsilon^{2/3})$-approximate SOSP.

Another line of work for escaping saddle points is to utilize the negative curvature finding (NCF), which can be combined with $\epsilon$-approximate first-order stationary point (FOSP) finding algorithms to find an $(\epsilon, \delta)$-approximate SOSP. The main task of NCF is to calculate the approximate smallest eigenvector of the Hessian for a given point. Classical methods for solving NCF like the power method and Oja's method need the computation of Hessian-vector products. Based on the fact the Hessian-vector product can be approximated by the finite difference between two gradients, [42, 4] proposed the FO NCF frameworks Neon+ and Neon2, respectively. In general, adding perturbations in the negative curvature direction can escape saddle points more efficiently than adding random perturbations by a factor of $\tilde{\mathcal{O}}(\text{poly}(\log d))$ in theory. Specifically, in the deterministic setting, CDHS [9] combined with Neon2 can find an $(\epsilon, \delta)$-approximate SOSP in gradient query complexity $\tilde{\mathcal{O}}(\log d/\epsilon^{1.75})$. Recently, the same result was achieved by a simple single-loop algorithm [44], which combined the techniques of perturbed accelerated gradient descent and accelerated negative curvature finding. In the online stochastic setting, the best gradient query complexity result $\tilde{\mathcal{O}}(1/\epsilon^3)$ is achieved by SPIDER-SFO$^+$ [13], which combined the near-optimal $\epsilon$-approximate FOSP finding algorithm SPIDER and the NCF framework Neon2 to find an $(\epsilon, \delta)$-approximate SOSP.

However, the gradient information is not always accessible. Many machine learning and deep learning applications often encounter situations where the calculation of explicit gradients is expensive or even infeasible, such as black-box adversarial attack on deep neural networks [36, 31, 10, 6, 40] and policy search in reinforcement learning [38, 11, 25]. Thus, zeroth-order (ZO) optimization, which uses function values to estimate the explicit gradients as an important gradient-based black-box method, is one of the best options for solving this type of ML/DL problem. A considerable body of work has shown that ZO algorithms based on gradient estimation have comparable convergence rates to their gradient-based counterparts. Although many gradient estimation-based ZO algorithms have been proposed in recent years, most of them focus on the performance of converging to FOSPs [34, 18, 21, 13], and only a few of them on SOSPs [23, 41, 30, 5].

As mentioned above, although there have been several works of finding local minima via ZO methods, they utilized the techniques of random perturbations [23, 41], random search [30], and cubic regularization [5], as shown in Table 1, which are not the most efficient ones of escaping saddle points as discussed before. Specifically, in the deterministic setting, [23] proposed the ZO perturbed

stochastic gradient (ZPSGD) method, which uses a batch of Gaussian smoothing based stochastic ZO gradient estimators and adds a random perturbation in each iteration. As a result, ZPSGD can find an $\epsilon$-approximate SOSP using $\tilde{\mathcal{O}}\left(d^2/\epsilon^5\right)$ function queries. [41] proposed the perturbed approximate gradient descent (PAGD) method which iteratively conducts the gradient descent steps by utilizing the forward difference version of the coordinate-wise gradient estimators until it reaches a point with a small gradient. Then, PAGD adds a uniform perturbation and continues the gradient descent steps. The total function queries of PAGD to find an $\epsilon$-approximate SOSP is $\tilde{\mathcal{O}}\left(d\log^4 d/\epsilon^2\right)$. Recently, [30] proposed the random search power iteration (RSPI) method, which alternately performs random search steps and power iteration steps. The power iteration step contains an inexact power iteration subroutine using only the ZO oracle to conduct the NCF, and the core idea is to use a finite difference approach to approximate the Hessian-vector product. In the stochastic setting, [5] proposed a zeroth-order stochastic cubic regularization newton (ZO-SCRN) method with function query complexity $\tilde{\mathcal{O}}\left(d/\epsilon^{7/2}\right)$ using Gaussian sampling-based gradient estimator and Hessian estimator. Unfortunately, each iteration of ZO-SCRN needs to solve a cubic minimization subproblem, which does not have a closed-form solution. Typically, inexact solvers for solving the cubic minimization subproblem need additional computations of the Hessian-vector product [1] or the gradient [7].

Thus, it is then natural to explore faster ZO negative curvature finding based algorithms to make escaping saddle points more efficient. To the best of our knowledge, negative curvature finding algorithms with access only to ZO oracle is still a vacancy in the stochastic setting. Inspired by the fact that the gradient can be approximated by the finite difference of function queries with high accuracy, a question is: *Can we turn FO NCF methods (especially the state-of-the-art Neon2) into ZO methods without increasing the iteration complexity and turn ZO algorithms of finding FOSPs into the ones of finding SOSPs?*

**Contributions.** We summarize our main contributions as follows:

- We give an affirmative answer to the above question. We propose two ZO negative curvature finding frameworks, which use only function queries and can detect whether there is a negative curvature direction at a given point $x$ on a smooth, Hessian-Lipschitz function $f : \mathbb{R}^d \to \mathbb{R}$ in offline deterministic and online stochastic settings, respectively.

- We apply the proposed frameworks to four ZO algorithms and prove that these ZO algorithms can converge to $(\epsilon, \delta)$-approximate SOSPs, which are ZO-GD, ZO-SGD, ZO-SCSG, and ZO-SPIDER.

- In the deterministic setting, compared with the classical setting where $\delta = \mathcal{O}(\sqrt{\epsilon})$ [22, 24, 23, 41], or the special case $\delta = \epsilon^{2/3}$ [30], our Theorem 4 is always not worse than other algorithms in Table 1. In the online stochastic setting, all of our algorithms don't need to solve the cubic subproblem as in ZO-SCRN and our Theorem 6 improves the best function query complexity by a factor of $\tilde{\mathcal{O}}(1/\sqrt{\epsilon})$.

## 2 Preliminaries

Throughout this paper, we use $\|\cdot\|$ to denote the Euclidean norm of a vector and the spectral norm of a matrix. We use $\tilde{\mathcal{O}}(\cdot)$ to hide the poly-logarithmic terms. For a given set $\mathcal{S}$ drawn from $[n] := \{1, 2, \ldots, n\}$, define $f_{\mathcal{S}}(\cdot) := \frac{1}{|\mathcal{S}|} \sum_{i \in \mathcal{S}} f_i(\cdot)$.

**Definition 1.** *For a twice differentiable nonconvex function* $f : \mathbb{R}^d \to \mathbb{R}$,

- *$f$ is $\ell$-Lipschitz smooth if $\forall x, y \in \mathbb{R}^d, \|\nabla f(x) - \nabla f(y)\| \le \ell \|x - y\|$.*

- *$f$ is $\rho$-Hessian Lipschitz if $\forall x, y \in \mathbb{R}^d, \|\nabla^2 f(x) - \nabla^2 f(y)\| \le \rho \|x - y\|$.*

**Definition 2.** *For a twice differentiable nonconvex function* $f : \mathbb{R}^d \to \mathbb{R}$, *we say*

- *$x \in \mathbb{R}^d$ is an $\epsilon$-approximate first-order stationary point if $\|\nabla f(x)\| \le \epsilon$.*

- *$x \in \mathbb{R}^d$ is an $(\epsilon, \delta)$-approximate second-order stationary point if $\|\nabla f(x)\| \le \epsilon, \nabla^2 f(x) \succeq -\delta \mathbf{I}$.*

We need the following assumptions which are standard in the literature of finding SOSPs [4, 13, 44].

**Assumption 1.** *We assume that $f(\cdot)$ in (1) satisfies:*

- $\Delta_f := f(x_0) - f(x^*) < \infty$ *where* $x^* := \arg\min_x f(x)$.

- *Each component function $f_i(x)$ is $\ell$-Lipschitz smooth and $\rho$-Hessian Lipschitz.*

- *(For online case only) The variance of the stochastic gradient is bounded: $\forall x \in \mathbb{R}^d$, $\mathbb{E}\|\nabla f_i(x) - \nabla f(x)\|^2 \le \sigma^2$.*

We'll also need the following more stringent assumption to get high-probability convergence results of ZO-SPIDER.

**Assumption 2.** *We assume that Assumption 1 holds, and in addition, the gradient of each component function $f_i(x)$ satisfies $\forall i, x \in \mathbb{R}^d$, $\|\nabla f_i(x) - \nabla f(x)\|^2 \le \sigma^2$.*

## 2.1 ZO Gradient Estimators

Given a smooth, Hessian Lipschitz function $f$, a central difference version of the deterministic coordinate-wise gradient estimator is defined by

$$\hat{\nabla}_{coord}f(x) = \sum_{i=1}^{d} \frac{f(x + \mu e_i) - f(x - \mu e_i)}{2\mu} e_i, \qquad \text{(CoordGradEst)}$$

where $e_i$ denotes a standard basis vector with $1$ at its $i$-th coordinate and $0$ otherwise; $\mu$ is the smoothing parameter, which is a sufficient small positive constant. A central difference version of the random gradient estimator is defined by

$$\hat{\nabla}_{rand}f(x) = d\frac{f(x + \mu u) - f(x - \mu u)}{2\mu} u, \qquad \text{(RandGradEst)}$$

where $u \in \mathbb{R}^d$ is a random direction drawn from a uniform distribution over the unit sphere; $\mu$ is the smoothing parameter, which is a sufficient small positive constant.

**Remark 1.** ***Deterministic vs. Random****: CoordGradEst needs $d$ times more function queries than RandGradEst. However, as will be discussed in section 4, it has a lower approximation error and thus can reduce the iteration complexity. **Central Difference vs. Forward Difference** (please refer to Appendix A.1): Under the assumption of Hessian Lipschitz, a smaller approximation error bound can be obtained by the central difference version of both CoordGradEst and RandGradEst.*

## 2.2 ZO Hessian-Vector Product Estimator

By the definition of derivative: $\nabla^2 f(x) \cdot v = \lim_{\mu \to 0} \frac{\nabla f(x + \mu v) - \nabla f(x)}{\mu}$, we have $\nabla^2 f(x) \cdot v$ can be approximated by the difference of two gradients $\nabla f(x+v) - \nabla f(x)$ for some $v$ with small magnitude. On the other hand, $\nabla f(x + v), \nabla f(x)$ can be approximated by $\hat{\nabla}_{coord}f(x + v), \hat{\nabla}_{coord}f(x)$ with high accuracy, respectively. Then the coordinate-wise Hessian-vector product estimator is defined by:

$$\mathcal{H}_f(x)v \triangleq \sum_{i=1}^{d} \frac{f(x + v + \mu e_i) - f(x + v - \mu e_i) + f(x - \mu e_i) - f(x + \mu e_i)}{2\mu} e_i. \qquad (2)$$

Note that we do not need to know the explicit representation of $\mathcal{H}_f(x)$. It is merely used as a notation for a virtual matrix and can be viewed as the Hessian $\nabla^2 f(x_0)$ with minor perturbations. As stated in the following lemma, the approximation error is efficiently upper bounded.

**Lemma 1.** *Assume that $f$ is $\rho$-Hessian Lipschitz, then for any smoothing parameter $\mu$ and $x \in \mathbb{R}^d$, we have*

$$\|\mathcal{H}_f(x)v - \nabla^2 f(x)v\| \le \rho \left( \|v\|^2/2 + \sqrt{d}\mu^2/3 \right). \qquad (3)$$

The ZO Hessian-vector product estimator was previously studied in [43, 30], but we provide a tighter bound than that in Lemma 6 in [30]. This is because we utilize properties of the central difference version of the coordinate-wise gradient estimator under the Hessian Lipschitz assumption. It is then directly concluded that, if $f(\cdot)$ is quadratic, we have $\rho = 0$ and $\|\mathcal{H}_f(x)v - \nabla^2 f(x)v\| = 0$.

# 3 Zeroth-Order Negative Curvature Finding

In this section, we introduce how to find the negative curvature direction near the saddle point using zeroth-order methods. Recently, based on the fact that the Hessian-vector product $\nabla^2 f(x) \cdot v$ can be approximated by $\nabla f(x+v) - \nabla f(x)$ with approximation error up to $\mathcal{O}(\|v\|^2)$, [4] proposed a FO framework named Neon2 that can replace the Hessian-vector product computations in NCF subroutine with gradient computations and thus can turn a FO algorithm for finding FOSPs into a FO algorithm for finding SOSPs. Enlightened by Neon2, we propose two zeroth-order NCF frameworks (*i.e.*, *ZO-NCF-Online* and *ZO-NCF-Deterministic*) using only function queries to solve nonconvex problems in the online stochastic setting and offline deterministic setting, respectively.

## 3.1 Stochastic Setting

In this subsection, we focus on solving the NCF problem with zeroth-order methods under the online stochastic setting and propose *ZO-NCF-Online*. Before introducing *ZO-NCF-Online*, we first introduce *ZO-NCF-Online-Weak* with weak confidence of $2/3$ for solving the NCF problem.

We summarize *ZO-NCF-Online-Weak* in Algorithm 1. Specifically, *ZO-NCF-Online-Weak* consists of at most $T = \mathcal{O}(\frac{\log^2 d}{\delta^2})$ iterations and works as follows: Given a detection point $x_0$, add a random perturbation with small magnitude $\sigma$ as the starting point. At the $t$-th iteration where $t = 1, \ldots, T$, set $\mu_t = \|x_t - x_0\|$ to be the smoothing parameter $\mu$ in (2). Then we keep updating $x_{t+1} = x_t - \eta \mathcal{H}_{f_i}(x_0)(x_t - x_0)$ where $\mathcal{H}_{f_i}(x_0)(x_t - x_0)$ is the ZO Hessian-vector product estimator and stops whenever $\|x_{t+1} - x_0\| \geq r$ or the maximum iteration number $T$ is reached. Thus as long as Algorithm 1 does not terminate, we have that the approximation error $\|\mathcal{H}_{f_i}(x_0)(x_t - x_0) - \nabla^2 f_i(x_0)(x_t - x_0)\|$ can be bounded by $\mathcal{O}(\sqrt{d}r^2)$ according to Lemma 1. Note that, although the error bound is poorer by a factor of $\mathcal{O}(\sqrt{d})$ as compared to $Neon_{weak}^{online}$ in [4] which used the difference of two gradients to approximate the Hessian-vector product and achieve an approximation error up to $\mathcal{O}(r^2)$, with our choice of $r$ in Algorithm 1, the error term is still efficiently upper bounded.

---

**Algorithm 1** ZO-NCF-Online-Weak $(f, x_0, \delta)$

---

1: $\eta \leftarrow \frac{\delta}{C_0^2 \ell^2 \log(100d)}, T \leftarrow \frac{C_0^2 \log(100d)}{\eta \delta}, \sigma \leftarrow \frac{\eta^2 \delta^3}{(100d)^{3C_0}\rho}, r \leftarrow (100d)^{C_0}\sigma$
2: $\xi \leftarrow \sigma \frac{\xi'}{\|\xi'\|}$, with $\xi' \sim \mathcal{N}(0, \mathbf{I})$
3: $x_1 \leftarrow x_0 + \xi$
4: **for** $t = 1, \ldots, T$ **do**
5:     $\mu_t \leftarrow \|x_t - x_0\|$
6:     $x_{t+1} = x_t - \eta \mathcal{H}_{f_i}(x_0)(x_t - x_0)$ with $\mu = \mu_t$ and $i \in [n]$
7:     **if** $\|x_{t+1} - x_0\| \geq r$ **then return** $v = \frac{x_s - x_0}{\|x_s - x_0\|}$ for a uniformly random $s \in [t]$

**Return** $v = \perp$

---

Other than the additional error term caused by ZO approximation, the motivation of *ZO-NCF-Online-Weak* is almost the same as $Neon_{weak}^{online}$. That is, under reasonable control of the approximation error of the Hessian-vector product, using the update rule of Oja's method [35] to approximately calculate the eigenvector corresponding to the minimum eigenvalue of $\nabla^2 f(x_0) = \frac{1}{n} \sum_{i=1}^{n} \nabla^2 f_i(x_0)$. Under similar analysis, we conclude that as long as the minimum eigenvalue of $\nabla^2 f(x_0)$ satisfies $\lambda_{min}(\nabla^2 f(x_0)) \leq -\delta$, *ZO-NCF-Online-Weak* will stop before $T$ and find a negative curvature direction that aligns well with the eigenvector corresponding to the minimum eigenvalue of $\nabla f^2(x_0)$. Then we have the following lemma:

**Lemma 2** (ZO-NCF-Online-Weak). *The output $v$ of Algorithm 1 satisfies: If $\lambda_{min}(\nabla^2 f(x_0)) \leq -\delta$, then with probability at least $2/3$, $v \neq \perp$ and $v^\mathsf{T} \nabla^2 f(x_0)v \leq -\frac{3}{4}\delta$.*

We summarize *ZO-NCF-Online* in Algorithm 2. Specifically, *ZO-NCF-Online* repeatedly calls *ZO-NCF-Online-Weak* for $\Theta(\log(1/p))$ times to boost the confidence of solving the NCF problem from $2/3$ to $1 - p$. We have the following results:

**Lemma 3.** *In the same setting as in Algorithm 2, define $z = \frac{1}{m} \sum_{j=1}^{m} v^\mathsf{T}(\mathcal{H}_{f_{i_j}}(x_0))v$. Then, if $\|v\| \leq \frac{\delta}{16d\rho}$ and $m = \Theta(\frac{\ell^2}{\delta^2})$, with probability at least $1 - p$, we have $\left| \frac{z}{\|v\|^2} - \frac{v^\mathsf{T} \nabla^2 f(x)v}{\|v\|^2} \right| \leq \frac{\delta}{4}$.*

---

**Algorithm 2** ZO-NCF-Online

---

**Input:** $f(\cdot) = \frac{1}{n}\sum_{i=1}^{n} f_i(\cdot)$, $x_0$, $\delta > 0$, $p \in (0,1]$.
1: **for** $j = 1, 2, \cdots, \Theta(\log(1/p))$ **do**
2: $\quad$ $v_j \leftarrow$ ZO-NCF-Online-Weak $(f, x_0, \delta)$
3: $\quad$ **if** $v_j \neq \perp$ **then**
4: $\qquad$ $m \leftarrow \Theta(\frac{\ell^2 \log(1/p)}{\delta^2})$, $v' \leftarrow \Theta(\frac{\delta}{d\rho})v_j$
5: $\qquad$ Draw $i_1, \ldots, i_m$ uniformly randomly from $[n]$
6: $\qquad$ $z_j = \frac{1}{m}\sum_{k=1}^{m} \frac{(v')^T \mathcal{H}_{f_{i_k}}(x_0)v'}{\|v'\|^2}$
7: $\qquad$ **if** $z_j \leq -\frac{3\delta}{4}$ **then return** $v = v_j$
**Return** $v = \perp$

---

**Theorem 1.** *Let $f(x) = \frac{1}{n}\sum_{i=1}^{n} f_i(x)$ where each $f_i$ is $\ell$-smooth and $\rho$-Hessian Lipschitz. For every point $x_0 \in \mathbb{R}^d$, every $\delta \in (0, \ell]$, the output of Algorithm 2 $v$ satisfies that, with probability at least $1 - p$: If $v = \perp$, then $\nabla^2 f(x_0) \succeq -\delta \mathbf{I}$; If $v \neq \perp$, then $\|v\| = 1$ and $v^T \nabla^2 f(x_0) v \leq -\frac{\delta}{2}$. The total function query complexity is*

$$\mathcal{O}\left(\frac{d \log^2(d/p)\ell^2}{\delta^2}\right).$$

### 3.2 Deterministic Setting

In this subsection, we focus on solving the NCF problem with zeroth-order methods under the offline deterministic setting and propose *ZO-NCF-Deterministic*. We summarize *ZO-NCF-Deterministic* in Algorithm 3. Since we want to compute the eigenvector corresponding to the most negative eigenvalue of $\nabla^2 f(x_0)$ approximately, one can convert it into an approximated top eigenvector computation problem of $\mathbf{M} := -\frac{1}{\ell}\nabla^2 f(x_0) + (1 - \frac{3\delta}{4\ell})\mathbf{I}$. This is because all eigenvalues of $\nabla^2 f(x_0)$ in $[-\frac{3\delta}{4}, \ell]$ will be mapped to eigenvalues of $\mathbf{M}$ in $[-1, 1]$, and all eigenvalues of $\nabla^2 f(x_0)$ smaller than $-\delta$ will be mapped to eigenvalues of $\mathbf{M}$ greater than $1 + \frac{\delta}{4\ell}$.

---

**Algorithm 3** ZO-NCF-Deterministic

---

**Input:** Function $f(\cdot)$, point $x_0$, negative curvature $\delta > 0$, confidence $p \in (0,1]$.
1: $T \leftarrow \frac{C_1^2 \log \frac{d}{p}\sqrt{\ell}}{\sqrt{\delta}}$, $\sigma \triangleq (d/p)^{-2C_1}\frac{\delta}{T^4\rho}$, $r \triangleq (d/p)^{C_1}\sigma$
2: $\xi \leftarrow \sigma \frac{\xi'}{\|\xi'\|}$, with $\xi' \sim \mathcal{N}(0, \mathbf{I})$
3: $x_1 \leftarrow x_0 + \xi$, $y_0 \leftarrow 0$, $y_1 \leftarrow \xi$
4: **for** $t = 1, \ldots, T$ **do**
5: $\quad$ $\mu_t = \|y_t\|$
6: $\quad$ $y_{t+1} = 2\mathcal{M}(y_t) - y_{t-1}$ where $\mathcal{M}(y) = (-\frac{1}{\ell}\mathcal{H}_f(x_0) + (1 - \frac{3\delta}{4\ell}))y$
7: $\quad$ $x_{t+1} = x_0 + y_{t+1} - \mathcal{M}(y_t)$
8: $\quad$ **if** $\|x_{t+1} - x_0\| \geq r$ **then return** $v = \frac{x_{t+1} - x_0}{\|x_{t+1} - x_0\|}$
**Return** $v = \perp$.

---

Similar to *ZO-NCF-Online-Weak*, *ZO-NCF-Deterministic* starts by adding a random perturbation $\xi$ to the detection point $x_0$. To find the negative curvature direction $v$ of $\nabla^2 f(x_0)$ such that $v^T \nabla^2 f(x_0)v \leq -\frac{\delta}{2}$, the classical power method which updates through $x_{T+1} = x_0 + \mathbf{M}^T \xi$ [30] will take $T \geq \tilde{\Omega}(\frac{\ell}{\delta})$ number of iterations since eigenvalues of $\mathbf{M}$ greater than $1 + \frac{\delta}{4\ell}$ grows in a speed $(1 + \delta/\ell)^T$. To reduce the iteration complexity $T$, we can replace the matrix polynomial $\mathbf{M}^T$ with the matrix Chebyshev polynomial $\mathcal{T}_T(\mathbf{M})$ and virtually update $x_{T+1} = x_0 + \mathcal{T}_T(\mathbf{M})\xi$.

**Definition 3.** *Chebyshev polynomial $\{\mathcal{T}_n(x)\}_{n \geq 0}$ of the first kind is*

$$\mathcal{T}_0(x) = 1, \quad \mathcal{T}_1(x) = x, \quad \mathcal{T}_{n+1}(x) = 2x \cdot \mathcal{T}_n(x) - \mathcal{T}_{n-1}(x),$$

*then it satisfies* $\mathcal{T}_t(x) = \begin{cases} \cos(n \arccos(x)), & x \in [-1, 1] \\ \frac{1}{2}[(x - \sqrt{x^2 - 1})^n + (x + \sqrt{x^2 - 1})^n], & x > 1 \end{cases}$.

In the matrix case, we have the so-called matrix Chebyshev polynomial $\mathcal{T}_t(\mathbf{M})$ [3], which satisfies: $\mathcal{T}_{t+1}(\mathbf{M})\xi = 2\mathbf{M}\mathcal{T}_t(\mathbf{M})\xi - \mathcal{T}_{t-1}(\mathbf{M})\xi$. Thus, eigenvalues of $\mathbf{M}$ greater than $1 + \frac{\delta}{4\ell}$ will grow to $\left(1 + \delta/4\ell + \sqrt{(\delta/4\ell)^2 + \delta/2\ell}\right)^T \approx \left(1 + \sqrt{\delta/\ell}\right)^T$, so we only need to choose $T \geq \sqrt{\ell/\delta}$.

On the other hand, since we only have access to the zeroth-order information, we need to stably compute the matrix Chebyshev polynomial. In algorithm 3, we set $\mu_t = \|y_t\|$ and use $\mathcal{M}(y_t) = (-\frac{1}{\ell}\mathcal{H}_f(x_0) + (1 - \frac{3\delta}{4\ell}))y_t$ to approximate $\mathbf{M}y_t$ with approximation error up to $\frac{2\rho\sqrt{d}rt}{\ell}\|y_t\|$. With proper choice of $r$, it allows us to use the inexact backward recurrence [3] to ensure a stable computation of matrix Chebyshev polynomial:

$$y_0 = 0, \quad y_1 = \xi, \quad y_{t+1} = 2\mathcal{M}(y_t) - y_{t-1}.$$

Then the output $x_{T+1} = x_0 + y_{T+1} - \mathcal{M}(y_T)$ is close to $x_0 + \mathcal{T}_T(\mathbf{M})\xi$ with a small approximation error. Finally, we have the following theorem:

**Theorem 2.** *Let $f(x) = \frac{1}{n}\sum_{i=1}^n f_i(x)$ where each $f_i$ is $\ell$-smooth and $\rho$-Hessian Lipschitz. For every point $x_0 \in \mathbb{R}^d$, every $\delta \in (0, \ell]$, the output of Algorithm 3 $v$ satisfies that, with probability at least $1 - p$: If $v = \perp$, then $\nabla^2 f(x_0) \succeq -\delta\mathbf{I}$; If $v \neq \perp$, then $\|v\| = 1$ and $v^\mathsf{T}\nabla^2 f(x_0)v \leq -\frac{\delta}{2}$. The function query complexity is*

$$\mathcal{O}(\frac{d\log\frac{d}{p}\sqrt{\ell}}{\sqrt{\delta}}).$$

## 4 Applications of Zeroth-Order Negative Curvature Finding

In this section, we focus on applying the zeroth-order negative curvature frameworks to the following ZO algorithms: ZO-GD, ZO-SGD, ZO-SCSG, and ZO-SPIDER. The following result shows that one can verify if a point $x$ is an $\epsilon$-approximate FOSP using CoordGradEst.

**Proposition 1.** *In the online setting, using CoordGradEst with a batch size of $\mathcal{O}\left(\left(\frac{128\sigma^2}{\epsilon^2} + 1\right)\log\frac{1}{p}\right)$ and smoothing parameter $\mu \leq \sqrt{\frac{3\epsilon}{4\rho\sqrt{d}}}$, we can verify with probability at least $1 - p$, either $\|\nabla f(x)\| \geq \epsilon/2$ or $\|\nabla f(x)\| \leq \epsilon$. In the deterministic setting, using once computation of CoordGradEst with smoothing parameter $\mu \leq \sqrt{\frac{3\epsilon}{2\rho\sqrt{d}}}$, we can verify with probability $1$, either $\|\nabla f(x)\| \geq \epsilon/2$ or $\|\nabla f(x)\| \leq \epsilon$.*

### 4.1 Applying Zeroth-Order Negative Curvature Finding to ZO-GD and ZO-SGD

We apply ZO-NCF-Online to ZO-SGD to turn it into a local minima finding algorithm, and propose ZO-SGD-NCF in Algorithm 4. At each iteration, we use a batch size of $\mathcal{O}\left(\frac{\sigma^2}{\epsilon^2}\log\left(\frac{2K}{p}\right)\right)$ Coord-GradEst to verify if $x_t$ is an $\epsilon$-approximate stationary point. If not, ZO-SGD-NCF either estimates the gradient $\nabla f_S(x_t) = \frac{1}{|S|}\sum_{i\in S}\nabla f_i(x_t)$ by CoordGradEst (**Option I**) or RandGradEst (**Option II**) with both mini-batch size $\mathcal{O}(\frac{\sigma^2}{\epsilon^2})$; If so, we call the ZO-NCF-Online subroutine. Then, If we find an approximate negative curvature direction $v$ around $x_t$, then we update $x_{t+1}$ by moving from $x_t$ in the direction $v$ with step-size $\delta/\rho$. We have the following theorem:

**Theorem 3.** *Under Assumption 1, we set $\mu_1 = \sqrt{\frac{3\epsilon}{2\rho\sqrt{d}}}$ and other parameters as follows,*

***Option I:*** $|S| = \max\{\frac{32\sigma^2}{\epsilon^2}, 1\}, K = \mathcal{O}(\frac{\rho^2\Delta_f}{\delta^3} + \frac{\ell\Delta_f}{\epsilon^2}), \eta = \frac{1}{4\ell}, \mu_2 = \sqrt{\frac{3\epsilon}{4\rho\sqrt{d}}};$

***Option II:*** $|S| = \max\{\frac{8\sigma^2}{\epsilon^2}, 1\}, K = \mathcal{O}(\frac{\rho^2\Delta_f}{\delta^3} + \frac{d\ell\Delta_f}{\epsilon^2}), \eta = \frac{1}{32d\ell}, \mu_2 = \min\left\{\sqrt{\frac{3\epsilon}{4\rho d}}, \frac{\epsilon}{32\sqrt{d\ell}}\right\},$

*where $\mu_1$ and $\mu_2$ are only used in Line 3 and Line 5 (or Line 6) of Algorithm 4, respectively. With probability at least $1 - p$, Algorithm 4 outputs an $(\epsilon, \delta)$-approximate local minimum in function query complexity*

***Option I:*** $\tilde{\mathcal{O}}(\frac{d\sigma^2\ell\Delta_f}{\epsilon^4} + \frac{d\sigma^2\rho^2\Delta_f}{\epsilon^2\delta^3} + \frac{d\ell^2\rho^2\Delta_f}{\delta^5});$ ***Option II:*** $\tilde{\mathcal{O}}(\frac{d^2\sigma^2\ell\Delta_f}{\epsilon^4} + \frac{d\sigma^2\rho^2\Delta_f}{\epsilon^2\delta^3} + \frac{d\ell^2\rho^2\Delta_f}{\delta^5}).$

---

**Algorithm 4** ZO-SGD-NCF

---

**Input:** Function $f$, starting point $x_0$, confidence $p \in (0,1)$, $\epsilon > 0$ and $\delta > 0$.

1: **for** $t = 0, \ldots, K-1$ **do**
2:     uniformly randomly choose a set $\mathcal{B}$ with batch size $\mathcal{O}(\frac{\sigma^2}{\epsilon^2}\log(2K/p))$
3:     **if** $\|\hat{\nabla}_{coord}f_{\mathcal{B}}(x_t)\| \geq \frac{3\epsilon}{4}$ **then**
4:         uniformly randomly choose $S \subseteq [n]$
5:         **Option I :** $x_{t+1} \leftarrow x_t - \eta\hat{\nabla}_{coord}f_S(x_t)$
6:         **Option II :** $x_{t+1} \leftarrow x_t - \eta\hat{\nabla}_{rand}f_S(x_t)$
7:     **else**
8:         $v \leftarrow$ ZO-NCF-Online $(f, x_t, \delta, \frac{p}{2K})$
9:         **if** $v = \perp$ **then return** $x_t$
10:       **else**   $x_{t+1} = x_t \pm \frac{\delta}{\rho}v$

---

**Algorithm 5** ZO-GD-NCF

---

**Input:** Function $f$, starting point $x_0$, confidence $p \in (0,1)$, $\epsilon > 0$ and $\delta > 0$.

1: **for** $t = 0, \ldots, K-1$ **do**
2:     **if** $\|\hat{\nabla}_{coord}f(x_t)\| \geq \frac{3\epsilon}{4}$ **then**
3:         **Option I :** $x_{t+1} \leftarrow x_t - \eta\hat{\nabla}_{coord}f(x_t)$
4:         **Option II :** $x_{t+1} \leftarrow x_t - \eta\hat{\nabla}_{rand}f(x_t)$
5:     **else**
6:         $v \leftarrow$ ZO-NCF-Deterministic $(f, x_t, \delta, \frac{p}{K})$
7:         **if** $v = \perp$ **then return** $x_t$
8:         **else**   $x_{t+1} = x_t \pm \frac{\delta}{\rho}v$

---

**Remark 2.** *Note that the dominant term of the function query complexity in **Option I** is $\tilde{\mathcal{O}}(\frac{d}{\epsilon^4})$, while in **Option II** is $\tilde{\mathcal{O}}(\frac{d^2}{\epsilon^4})$. This is because CoordGradEst has a lower approximation error and thus can reduce the iteration complexity by a factor of $d$. Then the function query complexity of **Option II** is dominated by evaluating the magnitude of the gradient (Line 3 in Algorithm 4).*

In the Deterministic setting, we apply ZO-NCF-Deterministic to ZO-GD to turn it into a local minima finding algorithm and propose ZO-GD-NCF in Algorithm 5. The update rule of ZO-GD-NCF is similar to that in ZO-SGD-NCF, the only difference is that we don't need to use mini-batch sampling of the stochastic gradient. Similarly, we have the following theorem:

**Theorem 4.** *Under Assumption 1, we set $\mu_1 = \sqrt{\frac{3\epsilon}{2\rho\sqrt{d}}}$ and other parameters as follows,*

$$\text{\textbf{\textit{Option I:}}} \; K = \mathcal{O}(\frac{\rho^2\Delta_f}{\delta^3} + \frac{\ell\Delta_f}{\epsilon^2}), \eta = \frac{1}{4\ell}, \mu_2 = \sqrt{\frac{3\epsilon}{4\rho\sqrt{d}}};$$

$$\text{\textbf{\textit{Option II:}}} \; K = \mathcal{O}(\frac{\rho^2\Delta_f}{\delta^3} + \frac{d\ell\Delta_f}{\epsilon^2}), \eta = \frac{1}{8d\ell}, \mu_2 = \min\left\{\sqrt{\frac{3\epsilon}{4\rho d}}, \frac{\epsilon}{16\sqrt{d}\ell}\right\},$$

*where $\mu_1$ and $\mu_2$ are only used in Line 2 and Line 3 (or Line 4) of Algorithm 5, respectively. With probability at least $1 - p$, Algorithm 5 outputs an $(\epsilon, \delta)$-approximate local minimum in function query complexity*

$$\text{\textbf{\textit{Option I:}}} \quad \tilde{\mathcal{O}}(\frac{d\ell\Delta_f}{\epsilon^2} + d\frac{\sqrt{\ell}}{\sqrt{\delta}}\frac{\rho^2\Delta_f}{\delta^3}); \quad \text{\textbf{\textit{Option II:}}} \quad \tilde{\mathcal{O}}(\frac{d^2\ell\Delta_f}{\epsilon^2} + d\frac{\sqrt{\ell}}{\sqrt{\delta}}\frac{\rho^2\Delta_f}{\delta^3}).$$

## 4.2 Applying Zeroth-Order Negative Curvature Finding to ZO-SCSG and ZO-SPIDER

In the stochastic setting, we can also apply the zeroth-order negative curvature finding to the variance reduction-based algorithms: SCSG [28] and SPIDER [13]. Due to space limitation, We defer the detailed discussions of these applications to Appendix E and F.

To apply ZO-NCF-Online to SCSG, we first propose a zeroth-order variant of the SCSG [28] method in Algorithm 6. At the beginning of the $j$-th epoch, we estimate the gradient $\nabla f_{\mathcal{I}_j}(\tilde{x}_{j-1})$ by

CoordGradEst over a batch sampling set $\mathcal{I}_j$ with size $B$. In the inner loop iterations, the stochastic gradient estimator $v_{k-1}^j$ is either constructed by CoordGradEst or by RandGradEst over a mini-batch sampling set $\mathcal{I}_{k-1}^j$ with size $b$. Then we apply ZO-NCF-Online to ZO-SCSG and propose the ZO-SCSG-NCF method (see Algorithm 7).

**Theorem 5** (informal, full version deferred to Appendix E)**.** *With probability at least $\frac{2}{3}$, for both **Option I** and **Option II**, Algorithm 7 outputs an $(\epsilon, \delta)$-approximate local minimum in function query complexity*

$$\tilde{\mathcal{O}}(d(\frac{\ell\Delta_f}{\epsilon^{\frac{4}{3}}\sigma^{\frac{2}{3}}} + \frac{\rho^2\Delta_f}{\delta^3})(\frac{\sigma^2}{\epsilon^2} + \frac{\ell^2}{\delta^2}) + d\frac{\ell\Delta_f}{\epsilon^2}\frac{\ell^2}{\delta^2}).$$

We apply ZO-NCF-Online to ZO-SPIDER to turn it into a local minima finding algorithm and propose ZO-SPIDER-NCF in Algorithm 8. As a by-product, we also propose a zeroth-order variant of the SPIDER method in Appendix G that can converge to an $\epsilon$-approximate FOSP with high probability rather than expectation. Using the same technique as in SPIDER-SFO$^+$ [13], that is, instead of moving in a large single step with size $\delta/\rho$ along the approximate negative curvature direction as in ZO-SGD-NCF and ZO-SCSG-NCF, we can split it into $\delta/(\rho\eta)$ equal length mini-steps with size $\eta$. As a result, we can maintain the SPIDER estimates and improve the so-called non-improvable coupling term $\frac{1}{\delta^3\epsilon^2}$ by a fact of $\delta$.

**Theorem 6** (informal, full version deferred to Appendix F)**.** *With probability at least $\frac{3}{4}$, Algorithm 8 outputs an $(\epsilon, \delta)$-approximate local minimum in function query complexity*

$$\tilde{\mathcal{O}}\left(d\left(\frac{\sigma\ell\Delta_f}{\epsilon^3} + \frac{\sigma\ell\rho\Delta_f}{\epsilon^2\delta^2} + \frac{\ell^2\rho\Delta_f}{\delta^3\epsilon} + \frac{\ell^2\rho^2\Delta_f}{\delta^5} + \frac{\sigma^2}{\epsilon^2} + \frac{\sigma\delta\ell}{\rho\epsilon^2} + \frac{\ell^2}{\delta^2}\right)\right).$$

**Remark 3.** *We can boost the confidence the of Theorem 5 and 6 to $1 - p$ by running $\log(1/p)$ copies of Algorithm 7 and 8.*

## 5 Numerical Experiments

**Octopus Function.** We first consider the octopus function proposed by Du et al. [12]. The octopus function has $2^d$ local optimum: $x^* = (\pm 4\tau, \ldots, \pm 4\tau)^\mathsf{T}$ and $2^d - 1$ saddle points:

$$(0, \ldots, 0)^\mathsf{T}, (\pm 4\tau, 0, \ldots, 0)^\mathsf{T}, \ldots, (\pm 4\tau, \ldots, \pm 4\tau, 0)^\mathsf{T}.$$

We compare ZO-GD-NCF, ZPSGD, PAGD, and RSPI on the octopus function with growing dimensions. The parameters corresponding to the octopus function are set with $\tau = e, L = e, \gamma = 1$. All algorithms are initialized at point $(0, \ldots, 0)^\mathsf{T}$, which is a strict saddle point and the one farthest from the optimal points among the $2^d - 1$ saddle points.

We set $\epsilon = 1e-4, \delta = \sqrt{\rho\epsilon}$ for all experiments and report the function value v.s. the number of function queries in Figure 1. For RSPI, we follow the hyperparameter update strategy as described in ([30], Appendix, Section F): We keep $\sigma_2$ constant and update $\sigma_1 = \rho_{\sigma_1}\sigma_1$ every $T_{\sigma_1}$ iterations. We conduct a grid search for $T_{\sigma_1}$ and $\rho_{\sigma_1}$.

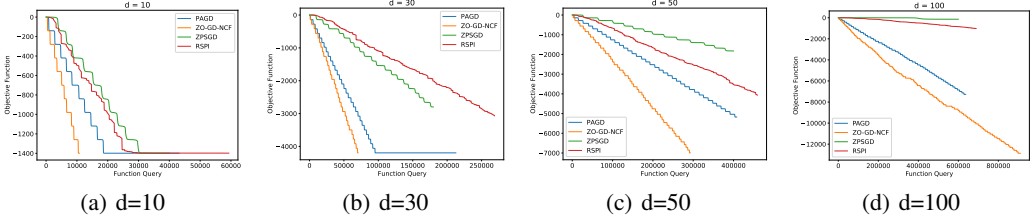

|  (a) d=10  |  (b) d=30  |  (c) d=50  |  (d) d=100  |

Figure 1: Performance of ZO-GD-NCF, ZPSGD, PAGD, and RSPI on the octopus function with growing dimensions.

The results in Figure 1 illustrate that all algorithms are able to escape saddle points. With the increase of the dimension of the octopus function, more function queries are needed for each algorithm to converge to the local minimum. Note that in all experiments, RSPI performs worse than PAGD and

ZO-GD-NCF. This is because RSPI is not a gradient based algorithm. Although it can efficiently escape from the saddle point using the negative curvature finding, it converges very slowly when the current point is far from the saddle point due to the random search.

We defer more experimental results to Appendix G.

## 6  Conclusion

In this paper, we analyse two types of ZO negative curvature finding frameworks, which can be used to find the negative curvature directions near a saddle point in the deterministic setting and stochastic setting, respectively. We apply the two frameworks to four ZO algorithms and analyse the complexities for converging to $(\epsilon, \delta)$-approximate SOSPs. Finally, we conduct several numerical experiments to verify the effectiveness of the proposed method in escaping saddle points.

As a future work, it would be interesting to study the (zeroth-order) unified negative curvature finding frameworks with generic analysis that can be applied to any FOSPs finding algorithms.

## Acknowledgments and Disclosure of Funding

The authors thank four anonymous reviewers for their constructive comments and suggestions. Bin Gu was partially supported by the National Natural Science Foundation of China under Grant 62076138.

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
