# OpenReview forum: "Zeroth-Order Negative Curvature Finding: Escaping Saddle Points  without Gradients"
_NeurIPS.cc/2022/Conference — NeurIPS 2022 Accept_

### Official Review · Reviewer_keP3 · 2022-07-08

**Rating:** 6
**Confidence:** 4
**Soundness:** 3 good
**Presentation:** 2 fair
**Contribution:** 3 good

**Summary:**

This work proposed zeroth-order negative curvature finding (NSF) algorithms for both deterministic and stochastic nonconvex minimization problem that finds $\epsilon$-SOSP with the same iteration complexity as the state-of-the-arts for finding $\epsilon$-FOSP, and improved the query complexity of existing algorithms for finding $\epsilon$-SOSP.

**Questions:**

(1) (Important) Line 5 of Algorithm 4 and Line 4 of Algorithm 7 use $\|\nabla f(x_t)\|\ge\frac{\epsilon}{2}$ while line 4 of Algorithm 5 uses $\|\widehat{\nabla}_{coord}f(x_t)\|\ge\frac{3\epsilon}{4}$, why do they differ? Due to proposition 1, I think you mean CoordGradEst can verify whether $\|\nabla f(x_t)\|\ge\frac{\epsilon}{2}$ with probability $1-p$, so we can always use CoordGradEst as in line 4 of Algorithm 5, yes? If not, the computation of the full gradient $\nabla f(x_t)$ in Algorithm 4 and 7 involves $n$ function queries and thus the complexity should involve $n$, which seems to eliminate the advantage of SGD for large $n$.

(2) (Important) Your Theorems 3 and 4 have probability at least $1-p$. Could you also guarantee that for Theorems 5 and 6 which only have probability at least 2/3 and 3/4 respectively? (e.g. maybe by repeating algorithms multiple times or changing hyper-parameters) How will you accordingly adjust complexity? This is to guarantee that the complexity comparison is fair.

(3) (important) The checklist said you have listed limitations. Where are they?

(4) In the research question in line 87, should not we focus on query complexity instead of iteration complexity?

(5) Line 130 said "Central Difference vs. Forward Difference (please refer to 131 Appendix B.1)", where is such Appendix B.1?

(6) Will the small $\mu$ in the denominator cause numerical error? If so, How to avoid that?

(7) I did not fully understand your Algorithms 1 and 2 until I read Theorem 1. I think it better to make the big picture more clear at the beginning of Section 3. For example, you might first describe the goal of NCF problem: to find the the negative curvature direction $v$ at point $x_0$ defined as $v^{\top}\nabla^2 f(x_0)v\le ??$ such that when $x$ is a nearly saddle point, going along this direction $v$ will escape saddle point $x$. Is my understanding correct? Also, do your Algorithms 1 and 2 differ from [4] only in that you use $\mathcal{H}_f(x)v$ instead of $\nabla f(x+v)-\nabla f(x)$ in [4]? You might make this more clear in the paper.

(8) In line 168, "Note that, although the error bound is poor by ... up to $\mathcal{O}(r^2)$". This sentence seems incomplete since the whole sentence lies in "although". For example, do you mean "although ..., our algorithm only requires function values while [4] requires both function values and gradient values."? Also, it is better to use "poorer" than "poor".

(9) In line 6 of Algorithm 2, you might use $\sum_{k=1}^m$ and $f_{i_k}$ to differentiate from $j$ in $z_j$.

(10) In line 196, $\lambda_i(M)=-\frac{1}{\ell}\lambda_i\big[\nabla^2 f(x_0)\big]+\big(1-\frac{3\delta}{4\ell}\big)\in [-1,1]$ implies $\lambda_i\big[\nabla^2 f(x_0)\big]\in \big[-\frac{3\delta}{4},2\ell-\frac{3\delta}{4}\big]$, yes?

(11) The intuition of Algorithm 3 looks unclear to me. For example, does $T_t(M)\xi$ correspond to $y_t$? How does $x_{t+1}=x_0+y_{t+1}-\mathcal{M}(y_t)$ in line 215 correspond to line 7 of algorithm 3? Also, why not directly applying Algorithms 1 and 2 to the deterministic case ($n=1$) by replacing $\mathcal{H}_{f_i}$ with $\mathcal{H}_f$?

(12) In Algorithms 4 and 5, the two options can be moved to inside "if" since we do not need to compute them when "else" holds.

(13) In Proposition 1, does $\mathcal{O}\left(\left(\frac{128 \sigma^{2}}{\epsilon^{2}}+1\right) \log \frac{1}{p}\right)$ mean the batchsize? Also, the statement "$\|\nabla f(x)\|\ge \epsilon/2$ or $\|\nabla f(x)\|\le \epsilon$" always holds. So I wonder: do you actually mean that we can verify whether $\|\nabla f(x)\|\ge \epsilon/2$, and similarly we can verify whether $\|\nabla f(x)\|\le \epsilon$? It is better to make it more clear.

(14) In all the Theorems, do the query complexities mean the number of queries for $f(x)$, or for $f_i(x)$?

(15) Line 3 of Algorithm 7 can be moved into "if $\|\nabla f(x_t)\|\ge \frac{\epsilon}{2}$" since we do not need to compute ZO-SCSG when "else" holds. Also, I think it will be more concise if you remove "for $j=1,...,T$ do" in Algorithm 6 since we only need to run it for 1 epoch.

(16) You might briefly introduce what is SCSG and SPIDER at the beginning of Sections 4.2 and 4.3. For example, you might say they are variance reduction techniques for non-convex finite-sum problems, and SPIDER has the near-optimal sample complexity, etc.

(17) In Algorithm 8, line 3 can be moved into "if $w_1\neq \bot$".

(18) What novel analysis techniques are used in your proof?

**Ethics Review Area:**

["I don’t know"]

**Limitations:**

The checklist said you have listed limitations. I did not find them.

**Strengths And Weaknesses:**

Originality: This work is a novel combination of zero-th order algorithms with NSF algorithms. The related work looks adequate and clear to me.

Quality: The submission is a complete work. The claims are well supported by theorems and experiments. I did not find out where limitations of this work are explicitly mentioned.

Clarity: The introduction is clearly written and well organized. There are many unclear points about algorithms and theorems as shown in my questions.

Significance: The results are important since it demonstrates that the proposed zero-th order algorithms can escape saddle point faster than the previous zero-th order algorithms.

---

> ### Author Response · Authors · 2022-08-02
> **Response to Reviewer keP3**
>
> Dear reviewer, thank you for taking the time to review our submission. We sincerely appreciate your valuable comments. Please find detailed responses to your questions below.
>
> Re to Q1: Thanks for pointing this out. Yes, Line 5 of Algorithm 4 and Line 4 of Algorithm 7 mean that we can use a batch of coordinate-wise gradient estimator to verify whether $\|\nabla f(x)\| \ge \epsilon/2$. In the revised version, we have rewritten these components to avoid misunderstandings.
>
> Re to Q2: Thanks for pointing this out. We can boost the confidence of Theorems 5 and 6 to $1-p$ by running $\log\frac{1}{p}$ copies of ZO-SCSG-NCF and ZO-SPIDER-NCF. So the total function query complexity of Theorems 5 and 6 will not change much since we use the notation $\tilde{\mathcal{O}}$ to hide the log factors. In the revised version, we will add the corresponding remarks to explain this.
>
> Re to Q3: We are sorry we didn't include the limitation in the first submission. In the revised version, we have added the limitation of this work in the conclusion. The main limitation of this work we think is that we don't give a unified analysis on how to use the zeorth-order negative curvature finding frameworks to any first-order stationary point finding ZO algorithms. This is still an open question and we may explore it in future work.
>
> Re to Q4: Yes, when it comes to the oracle complexity in zeroth-order optimization, we focus on the function query complexity; when it comes to the oracle complexity in first-order optimization, we focus on gradient query complexity. Direct comparison between the two oracle is unfair since the problem setting and accessible information are different. Thus, we can focus on the iteration complexity, which is related to the convergence rate of the optimization algorithm. Our motivation is to design zeroth-order algorithms with comparable convergence rates to the first-order methods and not increase the iteration complexity.
>
> Re to Q5: Sorry, it should be A.1 here.
>
> Re to Q6: Yes, in theory, we should set $\mu$ as small as possible to reduce the error term. However, this is not applicable in practice due to the numerical errors. To fill this gap, $\mu$ should be tuned empirically such that the error term is not too large, but large enough to avoid numerical errors.
>
> Re to Q7: Thanks for your suggestions on writing, we will consider it seriously in the revised version. Other than replacing $\nabla f(x+v) - \nabla f(x)$ with $\mathcal{H}_f(x)v$, we also dynamically set the smoothing parameter $\mu_t$ to be $\|x_t - x_0\|$. By doing so, the smoothing parameter is not always too small and will grow rapidly so the numerical stability is guaranteed to some extent.
>
> Re to Q8: In the revised version, we have rewritten this sentence to be "Note that, although the error bound is poorer by a factor of $\mathcal{O}(\sqrt{d})$ as compared to $Neon^{online}_{weak}$ in [4] which used the difference of two gradients to approximate the Hessian-vector product and achieve an approximation error up to $\mathcal{O}(r^2)$, with our choice of $r$ in Algorithm 1, the error term is still efficiently upper bounded."
>
> Re to Q9: Thank you for pointing this point out. In the revised version, we have fixed it.
>
> Re to Q10: Since $f$ is $\ell$-smooth, then we have all eigenvalues $\lambda_i(\nabla^2 f(x_0)) \in [-\ell, \ell]$. Since $\lambda_i(M) = -\frac{1}{\ell}\lambda_i(\nabla^2 f(x_0)) + (1 - \frac{3\delta}{4\ell})$, we have $\lambda_i(\nabla^2 f(x_0)) \in [\frac{-3\delta}{4}, \ell]$ implies $\lambda_i(M) \in [\frac{-3\delta}{4 \ell}, 1] \subset [-1,1]$.
>
> Re to Q11: The goal of Line 6 in algorithm 3 is to stably compute the approximate matrix Chebyshev polynomial $\mathcal{T}_t(M)$ since we have only access to the zeroth-order information, which is known as the inexact backward recurrence for stable computation of matrix Chebyshev polynomial (Section 6 in https://arxiv.org/pdf/1608.04773.pdf, appendix B.1 in https://arxiv.org/pdf/1711.06673.pdf). In our method, we prove that the zeroth-order Hessian-vector estimator also satisfies the precondition of the stable Chebyshev sum theorem (Theorem 6.4 in https://arxiv.org/pdf/1608.04773.pdf).
>
> > why not directly applying Algorithms 1 and 2 to the deterministic case by replacing $\mathcal{H}_{f_i}$ by $\mathcal{H}_f$
>
> This is because if we do it that way, the algorithm will be reduced to the classical power iteration method and the total iteration complexity will be $\Omega(\frac{\ell}{\delta})$, which is poorer than $\Omega(\sqrt{\frac{\ell}{\delta}})$ by computing the approximate matrix Chebyshev polynomial. We have discussed this point below the algorithm.

---

> > ### Author Response · Authors · 2022-08-02
> > **Additional Response**
> >
> > Re to Q12, Q15: Thank you for your suggestions. We will fix it in the revised version.
> >
> > Re to Q13: Yes, $\mathcal{O}\left( \left(\frac{128\sigma^2}{\epsilon^2}+1 \right) \log \frac{1}{p} \right)$ means the batch size. With our choice of the smoothing parameter $\mu$ we have $\| \nabla f(x) - \nabla_{coord} f(x) \| \le \frac{\epsilon}{4}$. So $\| \nabla_{coord} f(x)\| \ge \frac{3\epsilon}{4}$ implies $\|\nabla f(x)\| \ge \frac{\epsilon}{2}$ and $\|\hat{\nabla}_{coord} f(x)\| \le \frac{3\epsilon}{4}$ implies $\|\nabla f(x)\| \le \epsilon$.
> >
> > Re to Q14: Yes, the query complexity means the number of queries of $f(x)$ in the deterministic setting and $f_i(x)$ in the stochastic setting.
> >
> > Re to Q16: Thanks for your suggestion, in the revised version, we will add some descriptions of SCSG and SPIDER.
> >
> > Re to Q17: We don't agree with this statement because we move along the negative curvature direction $w_2$ by splitting it into $\delta/(\rho \eta)$ equal length mini-steps with size $\eta$ and in each mini-step, the direction $w_2$ cannot be changed. So this step should be put in the outer loop.
> >
> > Re to Q18: The novel analysis techniques are as follows:
> > - In Appendix A.1, we analyse the properties of zeroth-order gradient estimators $\nabla_{coord} f(x)$ and $\nabla_{rand} f(x)$ under the $\rho$-Hessian Lipschitz condition. Previous work only exploited the $\ell$-smoothness property of the gradient of f. We believe it is useful for studying the second-order convergence properties with zeroth-order methods. Also, we study the approximation error of the zeroth-order Hessian-vector product under the $\rho$-Hessian Lipschitz.
> > - In the applications of the zeroth-order negative curvature finding frameworks, since we need to verify if the norm of the current gradient is small enough, we propose Proposition 1. By using a batch of coordinate-wise gradient estimators with proper choice of the smoothing parameter, we can achieve this with high probability in the stochastic setting.
> > - We propose to use two different kinds of zeroth-order gradient estimators in ZO-SGD and ZO-SCSG for find second-order stationary points, which need novel and different analysis. To apply the ZO negative curvature finding algorithm to the SPIDER based algorithm, we first analyse the high probability convergence property of the zeroth-order SPIDER method for finding the first-order stationary points in Appendix F.1.

---

> > > ### Comment · Reviewer_keP3 · 2022-08-05
> > > **Reviewer keP3 agrees with the authors' reply and keeps rating 6**
> > >
> > > Reviewer keP3 agrees with the authors' reply and keeps rating 6.
> > > For Q11, I am curious why stochastic Algorithms 1-2 cannot use Chebyshev polynomial.
> > > Thank you.

---

> > > > ### Author Response · Authors · 2022-08-06
> > > > **Response to Q11**
> > > >
> > > > Thank you for pointing this out. Our Algorithm 1 can be seen as a zeroth-order version of the Oja's method, which is one of the most popular algorithm for PCA in stochastic setting and is theoretically optimal up to some log factors https://arxiv.org/pdf/1701.01722.pdf. A natural way to accelerate the Oja's method is to utilize the momentum scheme, which has a strong connection with the Chebyshev polynomial. An interesting work http://proceedings.mlr.press/v84/xu18a/xu18a.pdf studied the relationship of the momentum scheme and Chebyshev polynomial and concluded that "that adding momentum to a stochastic method like Oja’s does not always result in acceleration." Another recent work http://proceedings.mlr.press/v108/kim20e/kim20e.pdf also applied the variance reduced technique and heavy ball technique (momentum) to the Oja's method and achieved similar oracle complexity. In order not to increase the complexity of this paper, we don't consider these acceleration techiniques.

---

> > > > > ### Comment · Reviewer_keP3 · 2022-08-06
> > > > > **Thanks Response to Q11**
> > > > >
> > > > > In short, it seems that momentum with Chebyshev polynomial cannot improve convergence rate and complexity of stochastic optimization.
> > > > > Thanks for your clarification.

---

### Official Review · Reviewer_eedN · 2022-07-10

**Rating:** 7
**Confidence:** 3
**Soundness:** 3 good
**Presentation:** 3 good
**Contribution:** 4 excellent

**Summary:**

This work proposes a zeroth-order negative curvature framework (NCF) that escapes saddle points and converges to a second-order stationary point (SOSP). This framework can be applied to previous zeroth-order algorithms in order to converge to SOSPs. The theoretical results show that ZO-GD achieves similar convergence results to other deterministic ZO methods (with maybe a slightly better dependence on dimension $d$) when $\delta = \mathcal{O}(\sqrt{\epsilon})$. The stochastic results also achieve similar convergence results as other stochastic ZO methods except in the case $\delta = \mathcal{O}(\epsilon^{2/3})$.

**Questions:**

All of my questions are listed in the strengths and weaknesses above.

**Strengths And Weaknesses:**

To the best of my knowledge, this work seems to be quite novel. I have not seen any zeroth-order NCF methods, especially ones that obtain similar convergence results as its ZO peer methods. This is a big strength, and the literature review is in depth and helpful for the reader.

The theoretical analysis seems to be sound. I was unable to carefully examine all the proofs in the appendix. The paper seems to be only a theoretical contribution, which is fine but experimental results would be a boost to the overall work to show that this method is feasible and effective in practice (more on this later).

One major issue with this work is its cohesion: I felt that the work seems a bit thrown together. The theoretical results are lengthy, with many theorems/lemmas/corollarys/propositions not including detailed, descriptive, or informative remarks. The theoretical results should be described in a compact and easily digestable manner. In the current state, the results are quite sprawled in Section 4. The same is true for the algorithms. There are no detailed descriptions of Algorithms 4-8. This is problematic. Finally, there is no Conclusion section or any real end of the work. This makes the work feel like a work in progress.

The last issue is with the (lack of) experimental results. Many of the other ZO methods obtain similar convergence results and showcase their performance in practice. It would be beneficial to show if your ZO NCF method can perform better than a ZO random perturbation method. In my opinion this is something that is critically lacking in the current work.

Minor Issue:
1. Is Remark 2 supposed to come after Theorem 4? In the remark, it states that the "dominant term of the function query complexity in Option I is $\mathcal{\tilde{O}(d/\epsilon^2)}$". However, while true in Theorem 4, the dominant term seems to be $\mathcal{\tilde{O}(d/\epsilon^2 \delta^3)}$ in Theorem 3.

Overall, I feel that the development of a ZO-NCF method with rigorous theoretical analysis is a great contribution. However, I believe a lot of work still needs to be done to experimentally backup the convergence claims and make the paper cohesive.

I thank the authors for their submission and am looking forward to their response!

===========================================
Update after author rebuttal:

I had missed the experiments in the appendix, but the authors moved a single experiment into the main section (Section 5). Other experiments remain in the appendix. The authors also did a nice job in cleaning up the presentation of the work. For this, I have bumped up my overall score from a 5 to a 7 and presentation from a 2 to a 3. Great job!

**[R1]** Emmanouil-Vasileios Vlatakis-Gkaragkounis, Lampros Flokas, and Georgios Piliouras. Efficiently avoiding saddle points with zero order methods: No gradients required. Advances in Neural Information Processing Systems, 32, 2019.

---

> ### Author Response · Authors · 2022-08-02
> **Response to Reviewer eedN**
>
> Dear Reviewer, thank you for taking the time to review our submission. Please find detailed responses to your questions below. We hope that they will improve your opinion of our work and we kindly ask you to consider the possibility of raising your score.
>
> > I felt that the work seems a bit thrown together.
>
> Thank you for pointing this out. In the revised version, we have made some adjustments to the main paper based on your comments. Specifically, we summarize the application of the zero-order negative curvature finding framework to the algorithms zo-scsg and zo-spider in Section 4.2, and only keep the description of the algorithm and the informal theorem results. We defer the specific algorithms and formal theorems to the appendix.
>
> > The last issue is with the (lack of) experimental results.
>
> In fact, we did some experiments to verify that the proposed algorithm can effectively escape from saddle points, but due to space limitations, we deferred the experimental results to Appendix G. In the revised version, we add the numerical experiments section after section 4. In this section, we compare our ZO-GD-NCF method with ZPSGD, PAGD, and RSPI on the octopus function with growing dimensions. The experimental results show the effectiveness and efficiency of our method in escaping saddle points. The detailed parameter settings and other experimental results can be found in Appendix G.
>
> > Is Remark 2 supposed to come after Theorem 4?
>
> Thanks for pointing this out. Sorry, we made a small mistake here. This remark should come after Theorem 3. Since we have $\delta = \sqrt{\rho \epsilon}$ in the classical setting, the dominant term should be $\tilde{\mathcal{O}}(d/\epsilon^4)$ for Option I and $\tilde{\mathcal{O}}(d^2/\epsilon^4)$ for Option II.  In the revised version, we have fixed this mistake.

---

> > ### Comment · Reviewer_eedN · 2022-08-07
> > **Reviewer Follow-up**
> >
> > Dear Authors,
> >
> > Thank you for the thoughtful reply, and my apologies for the late response.
> >
> > I appreciate adding the experimental section into the paper (I somehow must have glossed over them in the appendix). The results look great (including the cubic regularization and regularized non-linear least squares examples in the appendix). Due to the inclusion of the experiments and the revised presentation, I will raise my score to a 7.

---

> > > ### Author Response · Authors · 2022-08-08
> > > **Thanks to Reviewer eedN**
> > >
> > > Thanks for reading our response and thanks for raising the score.

---

### Official Review · Reviewer_Sw24 · 2022-07-11

**Rating:** 6
**Confidence:** 3
**Soundness:** 3 good
**Presentation:** 2 fair
**Contribution:** 3 good

**Summary:**

This paper investigates escaping saddle points of nonconvex problems with zeroth order gradient descent method. A general framework is proposed to capture zeroth-order GD, SGD, and some other closely related algorithms. The main result is the convergence to approximate second order stationarities of these algorithms. The main idea is believed to be inspired by [18] and [47], but the results, to my best knowledge, is novel.

**Questions:**

As far as I have understood, a very related former work is [47]. Even the escaping saddle points results holds for algorithms that are not studied in [47], I would like to see the technical challenges of this paper compared to [47]. Part of the difficulty should come from the diversity and complexity of the algorithms considered in this paper, but anything else? I might have missed some points as reading the paper, it would be great if the authors are willing to provide some guidance.

**Limitations:**

Considered the choice of hyper parameters and the assumption of Lipschitz Hessian, current experiments is not sufficient for wide application of algorithms.

**Strengths And Weaknesses:**

Strength: The problem this paper addresses is well motivated, the writeup is easy to follow, and the novelty and contribution are easy to identified.

Weaknesses: As highlighted in the beginning of the paper, the main technique is to introduce a framework to capture differential ZO methods, but the main article has not focus on this unified framework. Instead of elaborating all the algorithms, the main idea and novelty should have been treated in the main to some extent.


The experimental results should have been presented somewhere in the main article. Reasons of doing this: the main difficulty of using perturbed methods (which have been proven escaping saddle points) is the setting of parameters, the experiments apparently have some hints on the effects of the parameters. Moreover, it is good to see experiments on the Octopus function since it is an important motivation of using any perturbed methods to escape saddle point efficiently instead of using much simpler algorithms like GD (and many first order methods) that has been proven not converging to saddle points either. Brief discussion on these results will make the paper more attractive.

---

> ### Author Response · Authors · 2022-08-02
> **Response to Reviewer Sw24**
>
> Dear Reviewer, thank you for taking the time to review our submission. We sincerely appreciate your valuable comments. Please find detailed responses to your questions below.
>
> > the main article has not focus on this unified framework.
>
> Thanks for pointing this out. Actually, how to apply the negative curvature finding frameworks to any FOSPs finding algorithm to turn it into a SOSPs finding algorithm is still an open question. The main difficulty is due to the fact that different first-order stationary point finding algorithms have different forms of descent lemmas. For example, in ZO-SGD with coordinate-wise gradient estimator we have $f(x_t) - \mathbb{E}[f(x_{t+1})] \ge \Omega(\|\nabla f(x)\|^2 - \frac{\epsilon^2 }{8})$ while in ZO-SCSG with coordinate-wise gradient estimator we have $f(x_t) - \mathbb{E}[f(x_{t+1})] \ge \Omega(\mathbb{E}\|\nabla f(x)\|^2 - \frac{\epsilon^2 }{8})$. Therefore, we need to analyze each algorithm separately. The study of the unified framework is beyond the scope of this paper, but we will explore this interesting research topic in future work.
>
> > The experimental results should have been presented somewhere in the main article.
>
> Thank you for your suggestion. In the revised version, we put the experiment of the octopus function in the section on numerical experiments.
>
> > I would like to see the technical challenges of this paper compared to [47]. Part of the difficulty should come from the diversity and complexity of the algorithms considered in this paper, but anything else?
>
> - [47] proposed the perturbed approximate gradient descent (PAGD) method for escaping saddle points, which approximates the gradient by the forward finite difference. The approximation error of this gradient estimator is bounded by $\|\hat{\nabla}_{coord} f(x) - \nabla f(x)\| \le \mathcal{O}(\ell \sqrt{d} \mu)$ when $f$ is $\ell$-smooth. In contrast, we approximate the gradient through the central finite difference and prove that the approximation error is bounded by $\mathcal{O}(\rho \sqrt{d} \mu^2)$ when $f$ is $\rho$-Hessian Lipschitz, which is a basic assumption in literature of analyzing the second-order convergence properties.
>
> - The main technique of PAGD for escaping saddle points is utilizing random perturbations. Specifically, when the norm of the current gradient is small, the algorithm adds a random perturbation in the region of a small ball $B_{\tilde{x}}(r)$ with radius r centered at current point $\tilde{x}$. It is proved in [1] that when $\tilde{x}$ is a saddle point, then by adding a random perturbation that is not in a small stuck region in $B_{\tilde{x}}(r)$, the function value will get further decrease after a number of steps. In contrast, the task of negative curvature finding is to directly find an approximate minimum eigenvector direction.
>
> [1] Chi Jin, Rong Ge, Praneeth Netrapalli, Sham M. Kakade, and Michael I. Jordan, How to escape saddle points efficiently.

---

### Official Review · Reviewer_2jjY · 2022-07-14

**Rating:** 7
**Confidence:** 3
**Soundness:** 3 good
**Presentation:** 4 excellent
**Contribution:** 3 good

**Summary:**

The paper addresses the problem of finding stationary points in non-convex settings that are not saddle points (i.e. satisfies second order optimality). It proposes two negative curvature finding (NCF) frameworks that only use zero-order oracles: one offline deterministic and the other online stochastic. The NCF frameworks provide two options of gradient estimation: a coordinate-based estimator and a randomized estimator. The paper then applies this framework to four algorithms and show that these algorithms converge to second-order stationary points. Convergence analysis gives complexity in terms of function queries.

**Questions:**

1. The authors describe situations when gradient calculations are expensive or infeasible (lines 55-56). Are there empirical results that their gradient-free methods outperform existing methods in these situations?
2. Experiments would also help to strengthen the claim that iteration complexity does not significantly increase. It would be good to have empirical results on the accuracy versus number of iterations needed, as well as accuracy versus training time, benchmarked against state of the art methods.
3. CoordEstGrad is calculated by summing across d finite differences calculation (where d is the dimensionality of the problem). This means that there is a dependency on d embedded within CoordEstGrad itself. Is this dependence on d included in the iteration complexity of the framework? For example, Remark 2 says "CoordGradEst has a lower approximation error and thus can reduce the iteration complexity by a fact[or] of d". Does this include the summation within the calculation of CoordGradEst? Similar question for Theorem 4 in the comparison of the complexity of CoordGradEst and that of RandCoordEst: does the O(d/epsilon2) vs O(d2/epsilon2) include the computation of CoordGradEst as well?
4. Some typos/ minor issues
 - Line 30. rho is undefined until line 108. It may be useful to have rho defined earlier.
 - Line 124. Although clear from context, mu is not defined until line 142.
 - Algorithm 1. What is C\_0? And what does ``with proper choice of r'' (line 171) mean? r is a calculated value in the algorithm based on d, C\_0 and sigma.
 -  Algorithm 3. What is C\_1?
 - Line 244. Typo? Should be "factor of d" instead of "fact of d".
5. Because of the similarity to the Neon2 paper, it could be useful to explicitly discuss the paper's contribution beyond what was done in Neon2. For example, which part of the theoretical analysis required different techniques than that used in the Neon2 paper?

**Limitations:**

This work does not appear to me to have negative social impact.

**Strengths And Weaknesses:**

Strengths:
 - Paper is well-written and logically organized. The notation is generally clear and easy to follow.
 - The paper addresses an important topic within optimization: escaping saddle points.

Weaknesses:
 - No experimental results.
 - Would like clarification on claim that their method turns first-order NCF methods into zero-order NCF methods without increasing the iteration complexity.

---

> ### Author Response · Authors · 2022-08-02
> **Response to Reviewer 2jjY**
>
> Dear Reviewer, thank you for taking the time to review our submission. We sincerely appreciate your valuable comments. Please find detailed responses to your questions below.
>
> >Q1: Are there empirical results that their gradient-free methods outperform existing methods in these situations?
>
> Yes, such empirical results can be found in the references we cite (Line 55-56). There are several application scenarios in machine learning where gradient-free methods are very useful, especially when the explicit gradient form is unavailable: Take the example of adversarial attack, the attacker has no access to parameters and the explicit formula of the model (like DNN) in the black-box scenario, while would like to maximize the loss regarding by adding imperceptible perturbation for a clean sample. Thus, zeroth-order optimization is a good tool for attacking black-box neural networks since we can approximate the gradient through the output of the model.
>
> >Q2: Experiments would also help to strengthen the claim that iteration complexity does not significantly increase.
>
> In the revised version, we add an experiment on the octopus function to compare the iteration performance between ZO-GD-NCF and Neon2+GD (see ). The results clearly show that our ZO-GD-NCD method will not significantly increase the iteration complexity compared to Neon2+GD.
>
> >Q3: Is this dependence on d included in the iteration complexity of the framework?
>
> The dimension d is not included in the iteration complexity (number of iterations) of the framework but in the oracle complexity (number of function queries). This is because the coordinate-wise gradient estimator can approximate the gradient with high accuracy (please refer to Lemma 8 in Appendix A.1). Then the ZO NCF framework can be seen as the FO NCF framework with minor perturbations and will not increase the number of iterations. In the literature of zeroth-order optimization, we mainly focus on the function query complexity. Since in each iteration we need to calculate the coordinate-wise gradient estimator, thus d will be included in the final function query complexity.
>
> >Q3: For example, Remark 2 says "CoordGradEst has a lower approximation error and thus can reduce the iteration complexity by a fact[or] of d". Does this include the summation within the calculation of CoordGradEst? Similar question for Theorem 4 in the comparison of the complexity of CoordGradEst and that of RandCoordEst: does the O(d/epsilon2) vs O(d2/epsilon2) include the computation of CoordGradEst as well?
>
> - Thanks for pointing this point out. We will start by explaining the relationship between approximation error and iteration complexity. In Lemma 8 and Lemma 9 in Appendix A.1, we derive that $\| \nabla_{coord} f(x) - \nabla f(x) \|^2 \le \frac{\rho^2 d \mu^4}{36}$ and $\mathbb{E} \| \nabla_{rand} f(x) - \nabla f_{\mu} (x) \|^2 \le \mathbb{E} \|\hat{\nabla}_{rand} f(x)\|^2 \le d \|\nabla f(x)\|^2 + \frac{\rho^2 d^2 \mu^4}{36}$, respectively. This means that the coordinate-wise gradient estimator always has a low approximation error while needing $\mathcal{O}(d)$ times more function queries in each iteration. On the other hand, we see that the random gradient estimator suffers from a high variance when the true gradient is large due to the term $d \|\nabla f(x)\|^2$. The results in algorithms 4 & 5 require $\mathcal{O}(d)$ times more iterations to ensure the convergence for Option II (Detail proofs can be found in Appendix D.1 and D.2). As a result, the total function queries for both Option I and Option II are almost the same in Line 5 and 6 of Algorithm ZO-SGD-NCF in the revised version (or Line 3 and 4 in ZO-GD-NCF).
>
> - In terms of the total function query complexity in Theorem 4, for both Option I and Option II, we need to evaluate the magnitude of the gradient by using the coordinate-wise gradient estimator in each iteration (Line 3 in ZO-SGD-NCF or Line 2 of ZO-GD-NCF, revised version). This results in $O(d/\epsilon^2)$ (Line 2 + Line 3) function query complexity for Option I and $O(d^2/\epsilon^2)$ (Line 2 + Line 4) function query complexity for Option II. To tackle this problem, a core idea is to reduce the frequency of evaluating the magnitude of the gradient. For example, in the stochastic setting, we can use the variance reduction-based algorithm like ZO-SCSG since we only need to evaluate the magnitude once in each epoch (i.e., the outer loop). Another way is to use the average random gradient estimator: $\frac{d}{q}\sum_{i=1}^q \frac{f(x+\mu u_i) - f(x - \mu u_i)}{2\mu} u_i$ in https://arxiv.org/abs/1805.10367. Then the term $d\|\nabla f(x)\|^2$ in the bound of variance of the random estimator will be reduced to $\frac{d}{q} \|\nabla f(x)\|^2$ and thus we can reduce the iteration complexity.

---

> > ### Author Response · Authors · 2022-08-02
> > **Additional Response**
> >
> > >Q4 typos/ minor issues
> >
> > Thanks for pointing these issues out.
> > - In the revised version, we have made $\rho$ and $\mu$ defined earlier (Line 28 and Line 126 in the revised version).
> > - $C_0$ in Algorithm 1 and $C_1$ in Algorithm 3 are sufficiently large constants. Proper choice of $r$ means as long as the value of $r$ is not too large, then the approximation error $\mathcal{O}(\sqrt{d}r^2)$ of the zeroth-order Hessian-vector estimator can be efficiently bounded. In the revised version, we have rewritten this sentence to be "with our choice of $r$ in Algorithm 1, the error term is still efficiently upper bounded."
> > - Yes, it should be "factor of d".
> >
> > >Q5: For example, which part of the theoretical analysis required different techniques than that used in the Neon2 paper?
> >
> > The novel analysis techniques are as follows:
> > - In Appendix A.1, we analyse the properties of zeroth-order gradient estimators $ \nabla_{coord} f(x) $ and $\nabla_{rand} f(x)$ under the $\rho$-Hessian Lipschitz condition. Previous work only exploited the $\ell$-smoothness property of the gradient of f. We believe it is useful for studying the second-order convergence properties with zeroth-order methods. Also, we study the approximation error of the zeroth-order Hessian-vector product under the $\rho$-Hessian Lipschitz.
> > - In the applications of the zeroth-order negative curvature finding frameworks, since we need to verify if the norm of the current gradient is small enough, we propose Proposition 1. By using a batch of coordinate-wise gradient estimators with proper choice of the smoothing parameter, we can achieve this with high probability in the stochastic setting.
> > - We propose to use two different kinds of zeroth-order gradient estimators in ZO-SGD and ZO-SCSG for finding second-order stationary points, which need novel and different analysis. To apply the ZO negative curvature finding algorithm to the SPIDER-based algorithm, we first analyse the high probability convergence property of the zeroth-order SPIDER method for finding the first-order stationary points in Appendix F.1.

---

> > ### Comment · Reviewer_2jjY · 2022-08-08
> > **Response after reading rebuttal**
> >
> > Thank you for the detailed replies to my questions and comments. I especially appreciate the detailed explanation on iteration complexity and the addition of empirical comparisons with Neon2+GD. Thank you for explaining the tradeoff between approximation error and iteration complexity, and that the dimensionality of the problem is included in the oracle call (function evaluation) itself. My point, however, is that iteration complexity itself may not capture the true cost of zero-order methods. And that the low approximation error of coordinate gradient estimate (vs that of random gradient estimator) is not free.
> >
> > Nevertheless, I overlooked the experiments that were in the appendix. It's much more clear now that the experiments are in the main body, and the practical advantage of the algorithm is also clearer. Fixing some typos and the general reorganization of the paper (e.g. adding a conclusion) makes the paper stronger. Thus, I increase my score from 6 to 7.

---

> > > ### Author Response · Authors · 2022-08-08
> > > **Thanks to Reviewer 2jjY**
> > >
> > > Thanks again for your further feedback and thanks for raising the score!

---

### Author Response · Authors · 2022-08-02
**General Response**

We thank all the reviewers for their constructive feedback! In our new submission, we made the following main revisions according to the reviewers' feedback:

- We have fixed all minor issues and typos in the revised version.

- We rewrite the pseudocode of the Algorithm 4, 5 & 7 according to suggestions from Reviewer keP3, which does not change the logic and complexity, just to increase the readability of the algorithms.

- We reorganized section 4. Specifically, we summarize the application of the zero-order negative curvature finding framework to the algorithms zo-scsg and zo-spider in Section 4.2, and only keep the description of the algorithm and the informal theorem results. We defer the specific algorithms and formal theorems to the appendix.

- Two reviewers mentioned that they didn't see the experimental results (actually, due to the space limitation, the experiments are already contained in Appendix G in our first submission), and one reviewer suggested that the experimental results should be presented somewhere in the main article. Thus, we add the numerical experiments section in the revised version. In this section, we compare our ZO-GD-NCF method with ZPSGD, PAGD, and RSPI on the octopus function with growing dimensions. The experimental results show the effectiveness and efficiency of our method in escaping saddle points.

- Finally, we add a conclusion section to summarize this work and discuss future research directions.

---

### Meta-Review · Area_Chair_N1nk · 2022-08-25

**Recommendation:** Accept
**Confidence:** Certain

**Metareview:**

This paper designs new algorithms for finding second order stationary points using only function value queries (0th order information). The main novelty is in designing two approaches for negative curvature finding. The new subroutines can be used in a wide range of algorithms for finding second order stationary points (most using first order information) and result in new 0th order algorithms with reasonable guarantees. The reviewers had some concerns but most are addressed in the response. In general the reviewers agree that this is a solid contribution to nonconvex optimization.

**Award:**

No

---

### Decision · Program_Chairs · 2022-09-14

Accept